# Whole-cell modeling of *E. coli* colonies enables quantification of single-cell heterogeneity in antibiotic responses

**Christopher J. Skalnik**[1]☯, **Sean Y. Cheah**[1]☯, **Mica Y. Yang**[1]☯, **Mattheus B. Wolff**[1], **Ryan K. Spangler**[1]¤, **Lee Talman**[2], **Jerry H. Morrison**[1], **Shayn M. Peirce**[2], **Eran Agmon**[1,3]*, **Markus W. Covert**[1]*

**1** Department of Bioengineering, Stanford University, Stanford, California, United States of America, **2** Department of Biomedical Engineering, University of Virginia, Charlottesville, Virginia, United States of America, **3** Center for Cell Analysis and Modeling, University of Connecticut School of Medicine, Farmington, Connecticut, United States of America

☯ These authors contributed equally to this work.
¤ Current address: Altos Labs, Redwood City, California, United States
* agmon@uchc.edu (EA); mcovert@stanford.edu (MWC)

**Data Availability Statement:** The full source code for the current version of the model is available in the vivarium-ecoli repository on GitHub (https://

## Abstract

Antibiotic resistance poses mounting risks to human health, as current antibiotics are losing efficacy against increasingly resistant pathogenic bacteria. Of particular concern is the emergence of multidrug-resistant strains, which has been rapid among Gram-negative bacteria such as *Escherichia coli*. A large body of work has established that antibiotic resistance mechanisms depend on phenotypic heterogeneity, which may be mediated by stochastic expression of antibiotic resistance genes. The link between such molecular-level expression and the population levels that result is complex and multi-scale. Therefore, to better understand antibiotic resistance, what is needed are new mechanistic models that reflect single-cell phenotypic dynamics together with population-level heterogeneity, as an integrated whole. In this work, we sought to bridge single-cell and population-scale modeling by building upon our previous experience in "whole-cell" modeling, an approach which integrates mathematical and mechanistic descriptions of biological processes to recapitulate the experimentally observed behaviors of entire cells. To extend whole-cell modeling to the "whole-colony" scale, we embedded multiple instances of a whole-cell *E. coli* model within a model of a dynamic spatial environment, allowing us to run large, parallelized simulations on the cloud that contained all the molecular detail of the previous whole-cell model and many interactive effects of a colony growing in a shared environment. The resulting simulations were used to explore the response of *E. coli* to two antibiotics with different mechanisms of action, tetracycline and ampicillin, enabling us to identify sub-generationally-expressed genes, such as the beta-lactamase ampC, which contributed greatly to dramatic cellular differences in steady-state periplasmic ampicillin and was a significant factor in determining cell survival.

github.com/CovertLab/vivarium-ecoli). Results can be reproduced by running on the version of the code archived at DOI 10.5281/zenodo.7709618. The data files used to generate all figures are deposited at DOI 10.5281/zenodo.7709450.

**Funding:** This work was supported by the Paul G. Allen Frontiers Group via the Allen Discovery Center at Stanford, as well as by a Sloan Foundation Matter-to-Life Award to M.W.C., an R01 award from the NLM of the National Institutes of Health under award number R01LM013229 to M.W.C. and from the NIGMS of the National Institutes of Health under award number R01GM140008 to M.W.C. This is publication #025 of the NSF Center for Chemical Currencies of a Microbial Planet (C-CoMP), an STC funded by the NSF. E.A. was supported by an F32 award from the NIGMS of the National Institutes of Health under award number F32GM137464, and M.Y.Y. was supported by a National Institutes of Health training grant under award number T32GM136568. The content is solely the responsibility of the authors and does not necessarily represent the official views of the National Institutes of Health, the National Science Foundation, the Sloan Foundation or the P.G. Allen Frontiers Group. The funders had no role in study design, data collection and analysis, decision to publish, or preparation of the manuscript.

**Competing interests:** The authors have declared that no competing interests exist.

## Author summary

Antibiotic-resistant bacteria pose a threat to human health, making current treatments for infection less effective or even obsolete. Computational modeling has been used to investigate phenomena related to antibiotic resistance at various scales, from diffusion of antibiotic molecules across cell barriers to the spread of resistance in hospitals. However, these models fail to capture phenomena that occur across multiple scales simultaneously. By combining multiple instances of a detailed mathematical model of individual *Escherichia coli* cells in a shared spatial environment, we were able to simulate bacterial colonies with single-cell detail of molecular mechanisms. We used this model to investigate the response of *E. coli* to two antibiotics with very different modes of action, evaluating how these responses were impacted by cell-to-cell variation in gene and protein expression. This work has implications for understanding emergent colony-level responses to antibiotics, and may offer a valuable approach to modeling colony-scale emergent phenomena more generally.

## Introduction

Antibiotic resistance poses mounting risks to human health, as current antibiotics are losing efficacy against increasingly resistant pathogenic bacteria. The prevalence of antibiotic-resistant phenotypes has long been recognized as a crisis [1,2]. Of particular concern is the emergence of multidrug-resistant strains, which has been rapid among Gram-negative bacteria such as *Escherichia coli* [2]. A large body of work has established that antibiotic resistance mechanisms depend on phenotypic heterogeneity, which may be mediated by stochastic expression of antibiotic resistance genes [3,4]. In addition to cellular heterogeneity, bacteria also demonstrate complex population-level behaviors such as bet hedging [3,5], quorum sensing [6], and long-range electrical signaling [7]. Some of these behaviors have already been shown to mediate the antibiotic response in large bacterial populations like biofilms [8]. Therefore, to better understand antibiotic resistance, new mechanistic models are needed that reflect single-cell phenotypic dynamics together with population-level heterogeneity, as an integrated whole.

To meet this need, systems biology and computational modeling efforts have quickly grown in scope and scale. Early studies employed a variety of mathematical approaches to quantify the transport of antibiotics across bacterial membranes, including ordinary differential equations [9] and graphical methods [10]. More recently, flux-balance analysis (FBA) models have been used to identify metabolic changes associated with antibiotic resistance [11,12]. Population dynamics [13] and agent-based [14] approaches have been used to characterize population-level antibiotic resistance mechanisms, such as slowed growth due to nutrient depletion in thick biofilms [15]. At the largest spatial scale, the bacteria themselves are abstracted away to facilitate modeling of large-scale phenomena including the transmission of antibiotic-resistant bacteria in hospitals [16] and the economic impact of antimicrobial resistance [17]. While all of these models have been successfully used to make predictions at their target scale, they are confined by their limited spatial resolution and are unable to generate meaningful conclusions about smaller or larger mechanisms of action.

In this work, we sought to bridge single-cell and population-scale modeling by building upon our previous experience in "whole-cell" modeling, an approach which integrates mathematical and mechanistic descriptions of biological processes to recapitulate the experimentally observed behaviors of entire cells [18]. In the decades since their conception, whole-cell

models have diversified greatly from their roots in ordinary differential equations [19], with two notable innovations coming in the form of constraint-based methods [20] and gene regulatory logic [21]. Whole-cell modeling efforts eventually culminated in a large-scale hybrid simulation model of *Mycoplasma genitalium* that accounted for the known functions of every annotated gene [18]. Since then, scientists have invested considerable effort into producing a whole-cell model of *Escherichia coli* [22]. The most recently published model includes the functions for 43% of the well-characterized genes, and encompasses 12 detailed submodels that collectively capture a wide range of cellular dynamics, including metabolism, chromosome replication, transcription, and translation [23]. This model was benchmarked against a variety of heterogeneous data, and successfully predicted the outcomes of experiments made after the simulations were performed [22].

To extend whole-cell modeling to the "whole-colony" scale, we utilized principles from agent-based modeling to embed multiple instances of a whole-cell *E. coli* model into a shared spatial environment. We accomplished this with the use of the Vivarium software library [24], which allowed us to run large, parallelized simulations on the cloud with all the molecular detail of the previous whole-cell model while adding many interactive effects of a colony growing in a shared dynamic environment. The resulting multiscale model was used to explore the response of *E. coli* to two antibiotics, tetracycline and ampicillin. Both antibiotics have seen extensive use as a treatment for infections, giving rise to many antibiotic-resistant strains of *E. coli* and other bacteria. In spite of this, tetracycline derivatives and ampicillin remain clinically relevant thanks in large part to research geared towards overcoming known resistance mechanisms [25,26]. By modeling the action of tetracycline and ampicillin in colonies of *E. coli*, we aimed to further our understanding of resistance to these classes of antibiotics and support future antibiotic development.

## Results

### Heterogeneity and interaction effects motivate whole-colony model

We were originally prompted to extend our cell-scale model to colonies in response to observations made while simulating the growth of individual bacterial cells. In particular, upon fitting transcription probabilities for each gene [22] to recapitulate experimentally measured transcript abundances, we noticed two qualitatively different patterns of gene expression (Fig 1A). The first pattern was expected (left panels, *ompF*); genes in this category exhibited what we termed exponential expression, with on average more than one transcription event per generation and stable, exponential protein production.

In contrast, the second dynamic gene expression pattern yielded significant variability in mRNA and protein expression and was seen in genes that were transcribed, on average, less than once per generation (right panels, *marR*). The protein products of such genes experienced sharp increases in number coinciding with each rare transcription event. These increases were typically followed by several generations of cellular protein loss due primarily to cell division (as opposed to decay). This sub-generational expression pattern has been experimentally observed for certain genes, including in *E. coli* [27,28], but we were surprised by the fact that more than half of all genes are expressed in this way (Fig 1B, outer circle) [22]. This finding could seem counter-intuitive, when considered from the perspective of a single cell: why wouldn't important gene products be expressed every generation, so that they would be available when needed? The answer can be seen from a whole-colony or population perspective: sub-generational expression allows the majority of cells to divert their resources towards more pressing cellular needs, while a minority hedges against the appearance of unanticipated environmental stressors or windfalls [3].

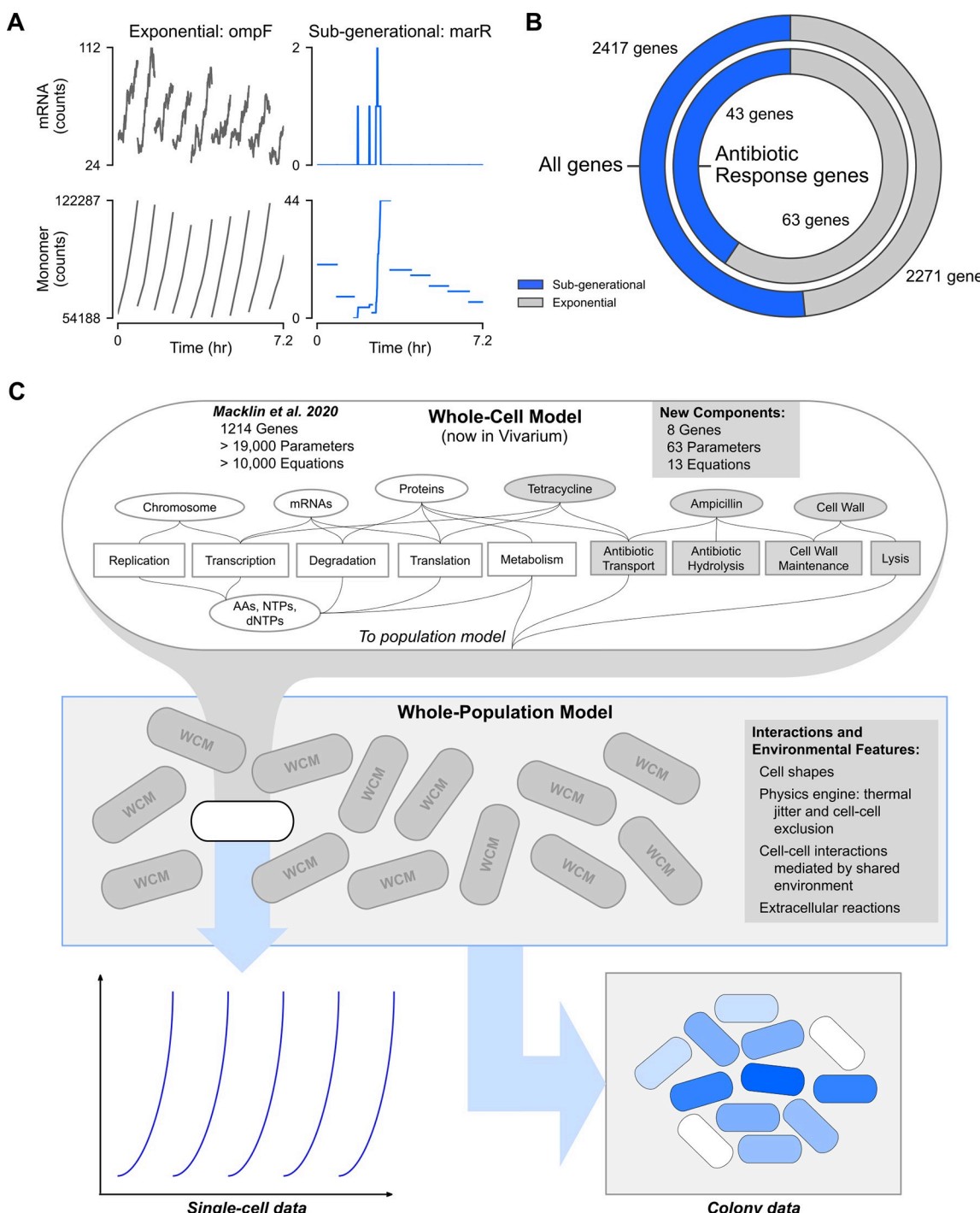

**Fig 1. Sub-generational gene expression of antibiotic response genes calls for a multi-scale model of the population-level antibiotic response.** (A) mRNA counts placed above corresponding monomer counts for a gene representative of "exponential" expression (left, *ompF*) and a gene representative of "sub-generational" expression on the right (right, *marR*). Data was taken from a representative lineage (cell 011001001 and its ancestors) in the simulated colony (seed 0) grown on minimal M9 medium supplemented with 1 mM glucose. (B) Proportion of all genes (outer ring) and antibiotic response genes (inner ring) that are predicted to be sub-generationally (blue) or exponentially (gray) expressed. Genes were considered sub-generationally expressed if <1 expression event per generation occurred on average in a baseline glucose simulation (seed 10000). (C) Schematic of the expanded *E. coli* model which can simulate both antibiotic responses and colony growth. The original whole-cell model [22] was supplemented with new functions for 8 genes, 63 new parameters, and 13 new equations. The expanded whole-cell model was then placed inside a spatial environment which supported propagation into colonies composed of many whole-cell model instances that share resources and physically interact. These whole-population models yielded rich time series data for single cells and spatial data for whole colonies.

In the context of this study, we were particularly intrigued by the finding that many known antibiotic resistance genes exhibited a sub-generational expression pattern (Fig 1B, inner circle). The highly stochastic expression of these genes makes them potential culprits behind the phenomenon antimicrobial heteroresistance, wherein subpopulations of isogenic bacteria vary greatly in susceptibility to specific antibiotics [29]. While most reports of heteroresistance focus on Gram-positive bacteria like *Staphylococcus aureus* [30–32], a subset of studies found similar behavior in Gram-negative species, including *E. coli* [33–37]. Thus, we decided to put our multi-cell model to use by implementing mechanisms for *E. coli*'s response to antibiotics, with the goal of studying the impacts of cellular heterogeneity on population-level phenotypes like heteroresistance.

One way to account for heterogeneity across a population would simply be to run a series of non-interacting, single-cell simulations that encompass an initial cell and all its eventual descendants. This approach would capture variability from stochastic gene expression and cell division, as well as any downstream effects. However, many groups have shown that the antibiotic response does not only involve phenotypic heterogeneity, but also intercellular interactions across the group. When susceptible *E. coli* were co-cultured with a resistant strain that composed only 6% of the total population, the entire colony gained protection against the beta-lactam cefamandole, indicating that high beta-lactamase activity from the resistant cells conferred protection to the susceptible cells [38]. In biofilms, high nutrient consumption by peripheral cells combined with reduced diffusion to interior regions led to starvation-induced growth arrest, directly increasing antibiotic tolerance [39,40]. Interestingly, tolerance to antibiotics and other toxins has been shown to depend on the age of a biofilm [41] and even the structural rigidity of the environment on which bacteria are grown [42]. With these and other interaction effects in mind, it quickly became clear that we would need to build a population-scale model to adequately simulate the response to antibiotics.

We began by reconstructing the existing *E. coli* whole-cell model [22] using Vivarium, a software tool that facilitates integrating diverse mechanistic models into a cohesive whole (Fig 1C) [24]. Vivarium helped us to add several new submodels to the whole-cell model, each of which encompasses an important aspect of *E. coli* single-cell response to ampicillin or tetracycline (gray elements in Fig 1C, top). The hierarchical structure of Vivarium-based simulations was then leveraged to combine multiple whole-cell models into a whole-population model (Fig 1C, middle). The resulting simulations were rich in heterogeneous, multi-scale data–spanning from the molecular activities and transformations of particular species over time, to the multicellular phenotypes that arise as one cell grows out into a more diverse population (Fig 1C, bottom). Moreover, our model retained information about the phylogenetic relationships between cells by assigning each daughter cell a unique barcode generated by appending a 0 or 1 to that of their mother cell, starting with 0 for the first cell, proceeding to 01 and 00 for its two daughters, etc. Full details of the model's construction and all the simulations shown here can be found in S1 Text; the complete source code can be found at our Github site (https://github.com/CovertLab/vivarium-ecoli).

## Simulated colonies exhibit phenotypic heterogeneity

Our first goal was to create a baseline for colony-scale simulations and ensure that our model behaved as expected. For this control, we ran 7.2-hour simulations initialized with a single cell growing in minimal M9 media supplemented with 1 mM glucose. Growth of the colony was simulated for roughly eight generations; a representative simulation is shown in Fig 2A.

As our simulations progressed, we observed that the number of cells in the colony roughly regularly doubled to form a circular, disorganized mass of cells (Fig 2A). As the size of the colony grew, glucose in the environment was depleted at an increasing rate, as would be expected

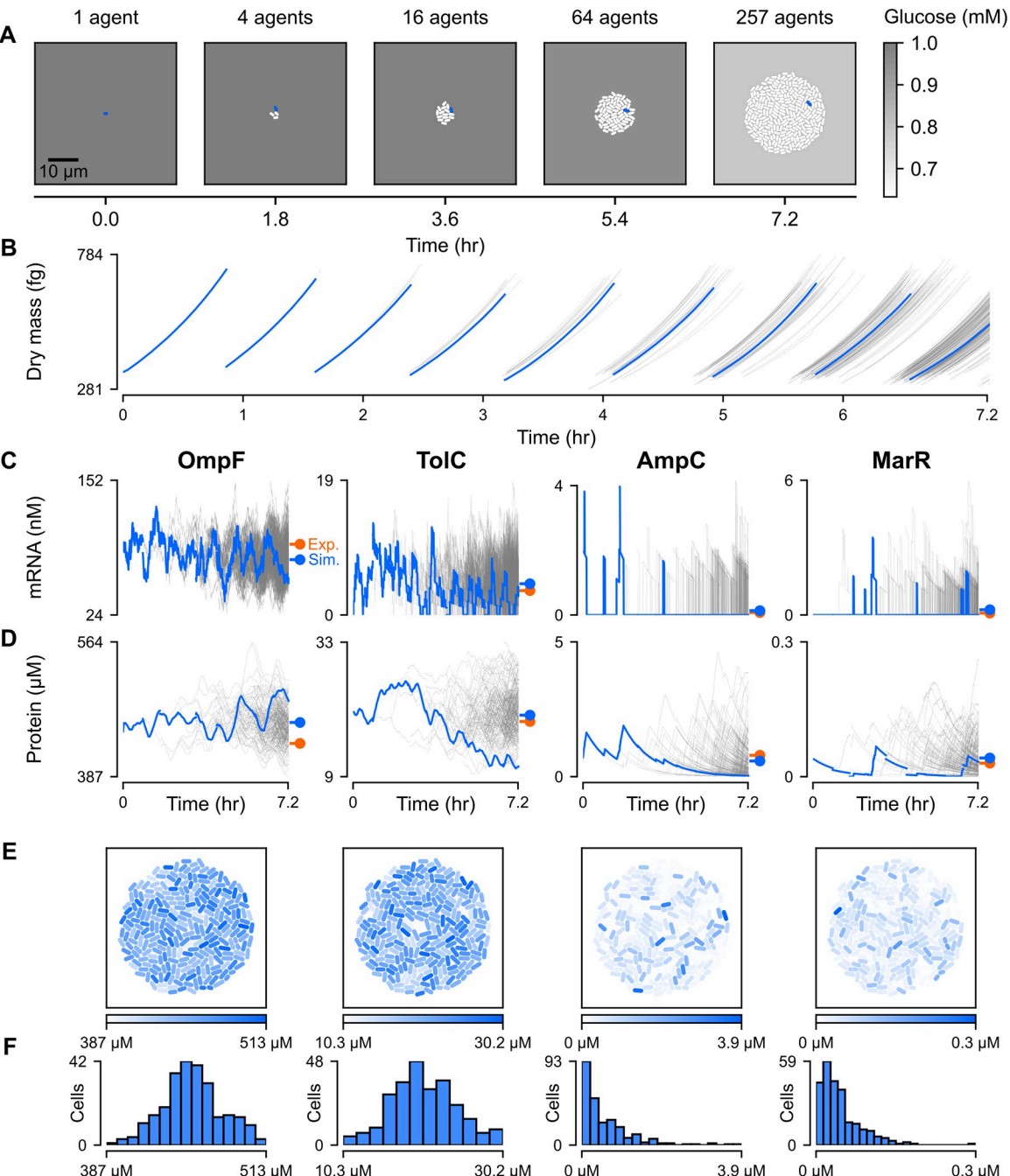

**Fig 2. Spatio-temporal heterogeneity in baseline gene expression.** (A) Snapshots of a simulated *E. coli* colony. The environmental glucose concentration is represented as varying shades of gray. The blue-colored cells are from a single representative cell lineage (cell 011000000 and its ancestors) and the time series data for that same lineage is also highlighted blue in panels B, C, and D. (B) Dry mass in femtograms for each cell in the colony over time. The blue lines are from the lineage whose members are blue in the snapshots of panel A. (C-D) Time series plots of mRNA concentration placed above corresponding plots of monomer concentration (both in nanomolar) for four genes relevant to antibiotic resistance. The blue lines are from the lineage whose members are blue in panel A. Colored markers indicate the average protein [51] and mRNA [52] concentrations from real (orange) and simulated cells (blue). (E-F) Colony snapshots taken at the end of the simulation with monomer concentrations (column labels same as in panel C) depicted using varying shades of blue. A histogram of the concentration distributions for all cells in the final colony is placed below its corresponding snapshot. *Note*: All data shown is from a simulation run with seed 10000 for about 7.2 hours (26000 seconds) on minimal M9 medium supplemented with 1 mM glucose.

from real growing colonies. Interestingly, in spite of significant cell-to-cell differences in glucose intake (coefficient of variation or CV = 0.15, S1A Fig), there was little variation in glucose concentrations between different regions in the environment, due primarily to the rapid time scales of diffusion (S1B Fig). This is consistent with prior experiments which indicated that oxygen was only completely depleted at the bottom of heterotrophic biofilms at least 200 μm in depth, far greater than the ≈15 μm radii of our largest microcolonies [43].

We then compared our simulated results to two large-scale datasets used to benchmark the original *E. coli* model [22]: cellular protein concentrations for 55% of predicted *E. coli* genes [44] and metabolic fluxes for a set of reactions in central carbon metabolism [45]. Compared to our original release of the whole-cell model, we observed roughly equal or better correlation between simulated and experimental results in both cases (S2 Fig). Next, we noted the measured doubling time of *E. coli* strain B/r (44 minutes), whose physiological and growth measurements were used to parameterize the model [46], was well within the distribution of doubling times for our model (50.89 ± 4.61 minutes, mean ± SD, S3A Fig). Interestingly, the slight amount of variability in doubling time was sufficient to visibly and progressively desynchronize cell division within the span of four generations (Fig 2B): cells entered Generation 4 (8 cells) with a standard deviation in start time of 2.57 minutes, which increased monotonically to 8.81 minutes for the start time of Generation 9 (S3B Fig).

We were also able to record the single-cellular expression of various genes across the colony. For example, we show the mRNA expression (Fig 2C) and protein expression (Fig 2D) traces over time, as well as the spatial (Fig 2E) and binned distributions (Fig 2F) of protein expression, for four key antibiotic resistance genes. From left to right, *ompF* codes for the primary outer membrane porin through which antibiotics enter the periplasm [47], *tolC* is translated into an outer membrane channel common to many drug efflux complexes [48], *ampC* is an endogenous beta-lactamase gene whose protein hydrolyzes beta-lactam antibiotics [49], and *marR* is a repressor of the multiple antibiotic resistance (*mar*) operon which confers low level antibiotic resistance in *E. coli* [50].

Like other exponentially expressed genes, mRNA expression was relatively steady and non-zero for the *ompF* and *tolC* genes. In the representative simulation shown in Fig 2, *ompF* mRNA remained non-zero in all cells for the entirety of the simulation, and the average cell had zero *tolC* mRNA for just 6.91% of its lifetime. This was in line with our expectations, given the multifunctional nature of the OmpF porin, which is known to facilitate nonspecific diffusion of small, non-antibiotic solutes [47], and the TolC channel, which forms efflux complexes that can operate on a variety of substrates [48]–suggesting that both gene products might be required by all cells, rather than part of an antibiotic bet-hedging strategy.

Conversely, *marR* and *ampC* exhibited key hallmarks of sub-generational expression: the average cell spent 91.4% of its lifetime with no *ampC* mRNA, and 86.0% of its lifetime with no *marR* mRNA. This included 59.0% of cells which had no *ampC* mRNA throughout their lifespans, and 41.8% which consistently had zero *marR* transcripts. Intuitively, the low expression of these two genes can be rationalized by the highly specific functions of their protein products: AmpC is a protective measure that is useful only in the presence of beta-lactams, while MarR controls the expression of an operon that responds primarily to antibiotic stress. In contrast, there were no cells that went through an entire cell cycle without transcription of the more broadly useful *ompF* or *tolC* genes. Thus, at the end of the simulation, counts of OmpF and TolC monomers varied far less (CV = 0.14, 0.24, respectively) than counts of AmpC and MarR monomers (CV = 1.2, 0.85, respectively) (Fig 2F). Interestingly, while MarR heterogeneity has been indirectly corroborated by measured inconsistency in expression of the *mar* operon activator MarA [3], less is known about the expression patterns of *ampC* and the implications of its potentially sub-generational expression, a topic we explore in a later section.

Our whole-colony approach also has the capacity to localize cells in physical space and characterize emergent patterns in spatial organization, which can otherwise be counter-intuitive. One might have expected, for instance, that cells with more recent common ancestors would necessarily be more proximal to one another in the colony. However, while daughter cells were always initially placed end-to-end following division, Brownian motion caused these cells to randomly drift apart from one another over time. We were surprised to find that while less closely related cells were more likely to be further apart in space (Spearman r = 0.45, p $\approx$ 0), the variance in phylogenetic relatedness, quantified as the number of edges in the shortest path between two cells in the binary phylogenetic tree, increased dramatically with distance (S4 Fig). Thus, while closely related cells were invariably close in space, being close in space did not guarantee that two cells were closely related (S4 Fig).

This inherent unpredictability in colony formation was reflected and amplified in snapshots of final protein concentrations (Fig 2E). Instead of distinct clusters of phenotypically similar cells, we observed a mixture of high expressing and low expressing cells, an observation that we confirmed using spatial autocorrelation analysis (S1 Table). Moreover, cells with a high average concentration for one of OmpF, TolC, MarR, or AmpC did not necessarily have a high average concentration for any of the others, further adding to cellular heterogeneity (S5 Fig). This indicated that, at least in the absence of coregulation (e.g. operon structure), simply having access to the same transcriptional and translational resources did not equate to universally high or low expression across distinct genes.

In sum, these results gave us confidence that the individual cell simulations in our colony were consistent with both experimental measurements and biological intuition, and also that the simulations as a whole exhibited heterogeneity across the colony for certain genes related to antibiotic resistance.

## Tetracycline uniformly inhibits growth of simulated colonies in a dose-dependent manner

Encouraged by our glucose simulations, we next wanted to determine how the whole-colony model would respond to tetracycline, a bacteriostatic antibiotic that binds to ribosomes and inhibits protein synthesis. Unlike bactericidal antibiotics (e.g. ampicillin), tetracycline does not directly cause cell death or lysis but instead slows colony growth (Fig 3A). The extent of this growth inhibition is dependent on several factors, primarily the amount of tetracycline that enters the cytoplasm and binds to ribosomes, as well as the expression of tetracycline resistance genes [53–55].

Net tetracycline flux into the *E. coli* cytoplasm is governed by both the rate of tetracycline diffusion across the outer and inner membranes, and by the rate of active efflux [9] (Fig 3B). Tetracycline is believed to cross the outer membrane through OmpF porins before passively diffusing across the inner membrane [9]. Once inside the cytoplasm, tetracycline is actively pumped out of the cell through the multidrug efflux pump AcrAB-TolC, which directly shuttles tetracycline from the cytoplasm, across both membranes, to the external environment [9]. In our model, the interplay between diffusive influx and active efflux was represented as a system of ordinary differential equations (ODEs), parameterized with previously determined membrane permeabilities and kinetic constants [9]. Notably, while permeability of tetracycline across the inner membrane was kept constant, outer membrane permeability was allowed to vary linearly in accordance with OmpF porin concentration, with lower and upper bounds taken from previously reported estimates [9]. The full equations and parameters are provided in S1 Text.

Once inside the cytoplasm, tetracycline can bind to a region on the 30S ribosomal subunit that is proximal to the A site of fully assembled ribosomes, preventing polypeptide elongation

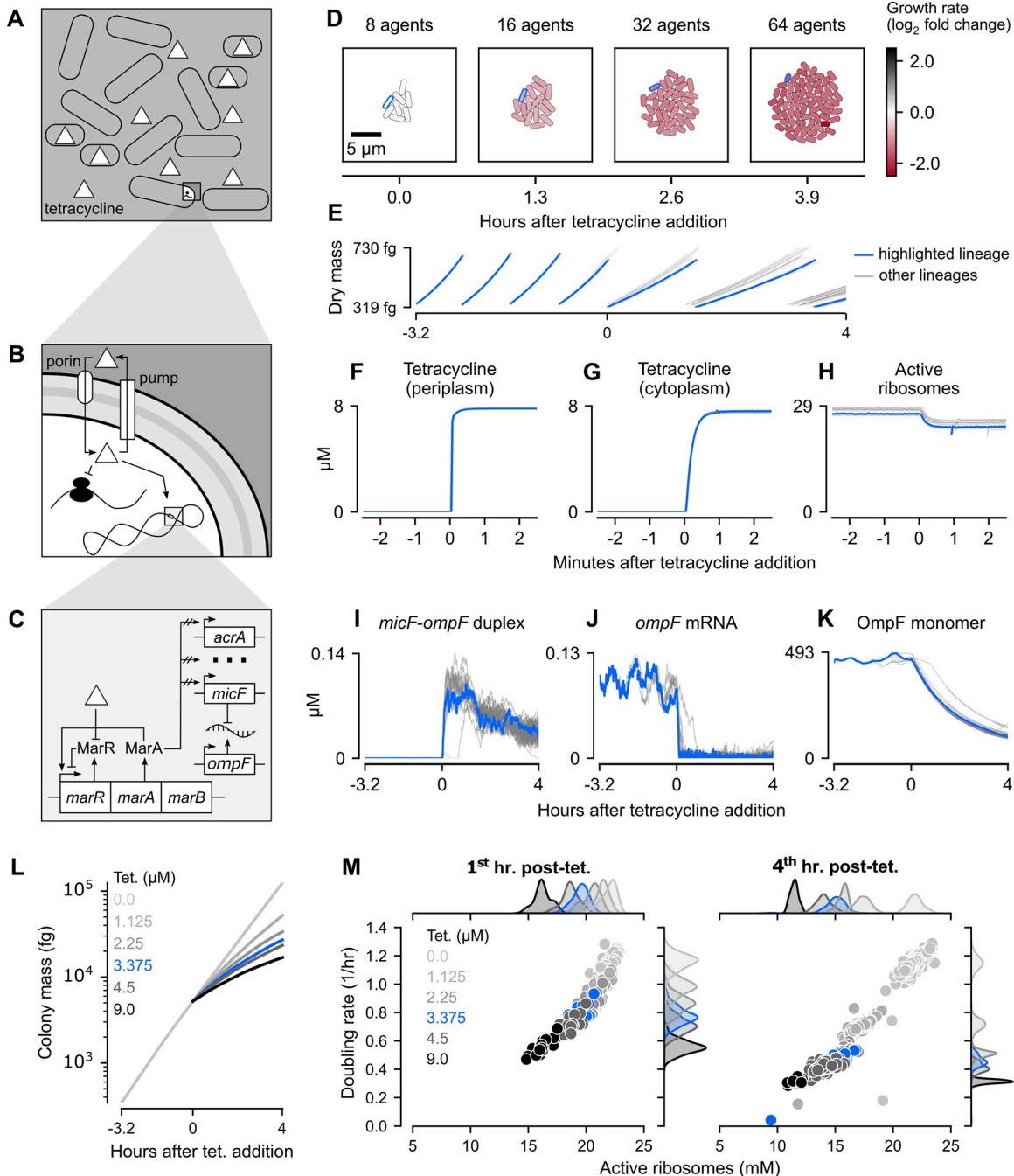

**Fig 3. Spatio-temporal dynamics of tetracycline action on *E. coli* colonies.** (A-C) Schematic of tetracycline response from diffusion at a colony scale to transport at a single-cell scale to regulation of gene expression at a molecular scale. (D) Snapshots of a representative colony simulation after addition of tetracycline at the MIC (1.5 mg/L). The color of each cell represents the $\log_2$ fold change of instantaneous growth rate compared to the average instantaneous growth rate of cells in a representative baseline glucose simulation. The cell lineage ending with cell 0011111 was outlined in blue and also plotted as blue traces in Panels E-K. (E) Per-cell dry mass starting with a single cell grown in M9 minimal medium with 1mM glucose and no tetracycline for about 3.2 hours (11550 seconds), at which point tetracycline was added at the MIC and the simulation continued for approximately 4 more hours (14550 seconds). (F-H) Per-cell concentration of, from left to right, periplasmic tetracycline, cytoplasmic tetracycline, and active ribosomes in the minutes surrounding tetracycline addition. (I-K) Per-cell concentration of, from left to right, micF-ompF RNA duplexes, ompF mRNA, and OmpF monomers for the entire 7.2-hour (26000 seconds) simulation. (L) Whole colony mass for representative simulations run across six different concentrations of tetracycline (MIC in blue). Data from these same simulations was also used to create Panel M. (M) Average active ribosome concentration and instantaneous doubling rate for all cells over the first (left) and fourth (right) hour of tetracycline exposure, with color corresponding to different tetracycline concentrations (MIC in blue). Marginal kernel density estimates were

normalized such that the area under each colored curve is 1. *Note*: All data shown is from simulations run with seed 0 initialized with a snapshot 3.2 hours (11550 seconds) into a baseline glucose simulation (also seed 0) before being continued for an additional four hours (14550 seconds) with varying levels of external tetracycline.

[56] (Fig 3B). For simplicity, we assumed that tetracycline competed with aminoacylated-tRNAs for occupancy of the A site and that the binding of both substrates were in chemical equilibrium. The binding constants for this equilibrium were selected from a range of previously published values [57–59] to yield a simulated dose-response curve comparable to *in vitro* measurements (S6A Fig) [60].

Exposure to tetracycline is also known to cause global changes in gene expression via induction of the *mar* operon, whose overexpression has been shown to confer resistance to a broad spectrum of antibiotics in *E. coli* [26,61]. The *mar* operon encodes the auto-repressor MarR, the auto- and general transcriptional activator MarA, and the small protein MarB of unclear function [62]. Tetracycline is believed to increase MarA expression by inactivating MarR, though the exact details of this mechanism are still unclear [63]. We approximated this by modeling MarR inactivation as a reversible reaction whose rate depends on the concentration of tetracycline in the cytoplasm, scaling the activity of MarA in proportion with the fraction of inactivated MarR. As its activity increased, more MarA was allowed to bind to the promoters of downstream target genes, transiently modifying their probability of transcription. The magnitude of this modulation was tuned for each gene to approximately recapitulate mRNA fold changes measured during tetracycline exposure (S6B Fig) [64]. Among the genes activated by MarA is the small antisense RNA *micF* [65]. This non-coding RNA post-transcriptionally inhibits OmpF porin production by forming untranslatable duplexes with *ompF* mRNA, a process assumed to be rapid and irreversible for simplicity in our model.

With these mechanisms represented in our model, we ran simulations on minimal M9 medium supplemented with 1 mM glucose and 1.5 mg/L (3.375 µM) tetracycline, well within the range of reported minimum inhibitory concentrations (MIC) for susceptible *E. coli* strains [9,66,67]. Each of these simulations was initialized with a saved snapshot (at t = 11550 seconds) of a non-antibiotic simulation and continued for an additional 14450 seconds (about 4 hours) of simulated time.

In all of our tetracycline simulations, we observed a significant decrease in instantaneous doubling rates (doubling of initial dry mass/hr) across the colony, with final averages nearly one-third (35.1 ± 4.52%, mean ± SD) that of cells in our untreated simulations (representative simulation in Fig 3D and 3E). The final colonies were also significantly smaller than those grown under non-inhibiting conditions, with an average of 63 cells compared to 254 cells in the baseline simulations.

Since cellular differences in OmpF monomer concentrations were minor at the instant of tetracycline addition, the variability in outer membrane permeability was similarly negligible (CV = 0.02 for both). As a result, tetracycline entered the periplasm at approximately the same rapid rate across all cells in the colony, yielding a CV in periplasmic tetracycline concentrations of less than 0.01 within 8 seconds of tetracycline addition (Fig 3F). We speculated that the decreased permeability of the inner membrane compared to the outer membrane, combined with active efflux by the AcrAB-TolC pump, might delay equilibration of tetracycline in the cytoplasm compared to the periplasm. Indeed, taking the equilibration time to be the first instant at which tetracycline changed by less than 0.05 nM/second, the periplasmic concentration reached steady-state in 78 seconds whereas the cytoplasmic concentration equilibrated in 102 seconds. Surprisingly, the substantial variability in the periplasmic AcrAB-TolC concentrations at the instant of tetracycline addition (CV = 0.24) did not translate to differences in

the equilibration time of cytoplasmic tetracycline, indicating that endogenous efflux was too slow to impact net influx at the MIC (Fig 3G). While active ribosome concentrations did decrease after tetracycline addition, there was minimal variability both at the moment of tetracycline addition and two minutes after (CV = 0.03 and 0.04, respectively; Fig 3H). Given the tight coupling between optimal growth rate and ribosome synthesis [68], this explains the relatively uniform reduction in growth rate seen as early as the second snapshot of Fig 3D.

Having observed the relatively rapid speed at which tetracycline is able to enter the cell and inhibit protein synthesis, we wondered whether the induction of the *mar* regulon would help cells develop meaningful resistance over time. In a representative simulation, *micF-ompF* duplexes reached a mean cytoplasmic concentration of 0.078 μM in the first ten minutes of tetracycline exposure (Fig 3I). Concomitantly, most cells experienced a steep decrease in translatable *ompF* mRNAs, the average concentration of which decayed to half its value at the instant of tetracycline addition within 5.23 minutes (Fig 3J). Despite this rapid and extreme reduction in mRNA levels, the average concentration of OmpF protein monomers decayed over a much longer time scale, taking 1.54 hours to reach half its value at the instant tetracycline introduction (Fig 3K). As a result, the average outer membrane permeability for tetracycline decreased from 97.4 ± 1.91 nm/s (mean ± SD) at the instant of tetracycline addition to 21.0 ± 2.60 nm/s (mean ± SD) four hours later (S7A Fig).

Since decreased outer membrane permeability directly limits the rate of tetracycline influx, we initially thought that active efflux may eventually overcome influx and either reduce or completely expel internal tetracycline. Contrary to our expectations, the observed 4.6-fold decline in outer membrane permeability was accompanied not by a decrease, but rather a slight but significant increase in the steady-state concentration of tetracycline in the cytoplasm (S7B Fig). This increase was accompanied by a significant decrease in the periplasmic concentration of the AcrAB-TolC efflux pump, suggesting that reduced efflux may be the underlying cause of this increased accumulation (S7C Fig). In both our model and in prior experiments, tetracycline exposure induced much more substantial upregulation of both *acrA* and *acrB* than *tolC* (S6B Fig) [64]. This disproportionate gene regulation, combined with inhibition of protein synthesis, gave rise to the observed reduction in AcrAB-TolC concentrations. Indeed, at sub-MIC levels of tetracycline, the balance between mRNA upregulation and inhibition of protein synthesis shifted, resulting in more gradual declines in the concentration of AcrAB-TolC (S7C Fig). Conversely, at tetracycline concentrations above the MIC, the concentration of AcrAB-TolC did not decrease more quickly than it did at the MIC, indicating that an equilibrium had been reached between mRNA and protein expression.

Globally, our model exhibited a wide dynamic range in its response to varying levels of external tetracycline (Fig 3L). Notably, while the MIC (highlighted in blue) yielded a pronounced reduction in colony mass after four hours, it did not completely suppress growth, and higher concentrations of tetracycline had increasingly potent inhibitory effects. Since the MIC is typically measured as the minimum concentration of an antibiotic required to prevent visible growth after overnight incubation [69], we speculated that *E. coli* are likely able to survive at the MIC but simply grow too slowly to generate visible growth after the usual incubation period. This hypothesis is qualitatively corroborated by experimental growth curves measured under different tetracycline concentrations, in which higher concentrations resulted in increasingly long lag periods where no growth was registered by a spectrophotometer [70].

Interestingly, the strength of growth inhibition by tetracycline increased with time, visible as a slight curvature in each of the log-scaled colony mass traces for a wide range of tetracycline concentrations (Fig 3L). In the first hour of tetracycline exposure, simulated cells exposed to higher tetracycline concentrations had, on average, lower active ribosome concentrations and lower growth rates, as expected (Fig 3M, left). By the fourth hour of tetracycline exposure,

however, the concentration of active ribosomes had almost universally decreased even further, deflating doubling rates and confirming that inhibition had increased over time (Fig 3M, right). The inhibition of protein synthesis by tetracycline had spawned a positive feedback loop in which reduced production of RNA polymerases and ribosomes progressively diminished RNA and protein synthesis (S8 Fig).

Taken together, our simulations in tetracycline-supplemented media revealed that inhibition of growth at a colony level may naturally compound over time due to positive feedback at a transcriptional and translational level, demonstrating the utility of our uniquely multi-scale approach.

## Ampicillin selectively kills simulated cells with low beta-lactamase concentrations

Next, we considered ampicillin, a bactericidal antibiotic that inhibits the activity of the penicillin-binding proteins (PBPs) which are responsible for cell wall synthesis and integrity [71–74] (Fig 4B). The presence of ampicillin and other beta-lactams in the periplasm interferes with the ability of PBPs to cross-link peptidoglycan strands in the cell wall, leading to accumulation of damage [75,76]. Critically damaged cells then lyse and release their internal beta-lactamase into the environment, where it continues to hydrolyze and inactivate extracellular ampicillin (Fig 4A). To capture the accumulation of damage and eventual lysis induced by ampicillin, we therefore needed to model transport of ampicillin into and out of the periplasm, binding and inactivation of PBPs by ampicillin, cell wall damage accumulating as a function of ampicillin interference with cell wall synthesis, lysis triggering the removal of a cell from the simulation, and release of beta-lactamase into the environment's reaction-diffusion system.

Transport of ampicillin into the cell was modeled as a system of differential equations that included not only diffusion and active efflux through AcrAB-TolC, but also hydrolysis by beta-lactamase (Fig 4B). Influx of ampicillin across the outer membrane was modeled using Fick's first law of diffusion. Active efflux through AcrAB-TolC was modeled using a Hill equation parameterized by experimentally determined $K_M$ and $v_{max}$ values [77]. Hydrolysis was likewise modeled using a Hill equation parameterized with $v_{max}$, $K_M$, and Hill coefficient values from literature [78]. Unlike tetracycline, ampicillin diffusion across the inner cytoplasmic membrane has been shown to be insignificant [77] and, as such, was not considered in our model. As before, the relevant parameters are included in S1 Text.

Once inside the periplasm, ampicillin covalently inactivated PBPs according to a Hill equation model. We focused on PBP1A and PBP1B, since these are the major PBPs responsible for elongation and crosslinking of cell wall peptidoglycans [71]. We parameterized the Hill equation model using an experimentally measured $IC_{50}$ for inhibition of PBP1B transpeptidase activity by ampicillin [79]. To our knowledge, no such value directly measuring the effect of ampicillin on transpeptidase activity has been reported for PBP1A. However, ampicillin binding affinities for PBP1A and PBP1B have been measured as being within one order of magnitude [74,80]. Assuming that the ability of ampicillin to inhibit transpeptidation is proportional to its binding affinity, we used this range of literature values to estimate an $IC_{50}$ for ampicillin inhibition of PBP1A transpeptidase activity that was within one order of magnitude of that measured for PBP1B and also resulted in death in simulations of single cells exposed to the ampicillin at the MIC.

PBPs bound by ampicillin were considered unable to contribute cross-linked peptidoglycans to our coarse-grained cell wall model. We represented the cell wall as a 2D lattice on the surface of a cylindrical shell, guided by prior measurements which have determined that the *E. coli* sacculus is mostly or completely a single layer [81–83]. Lattice positions represented the

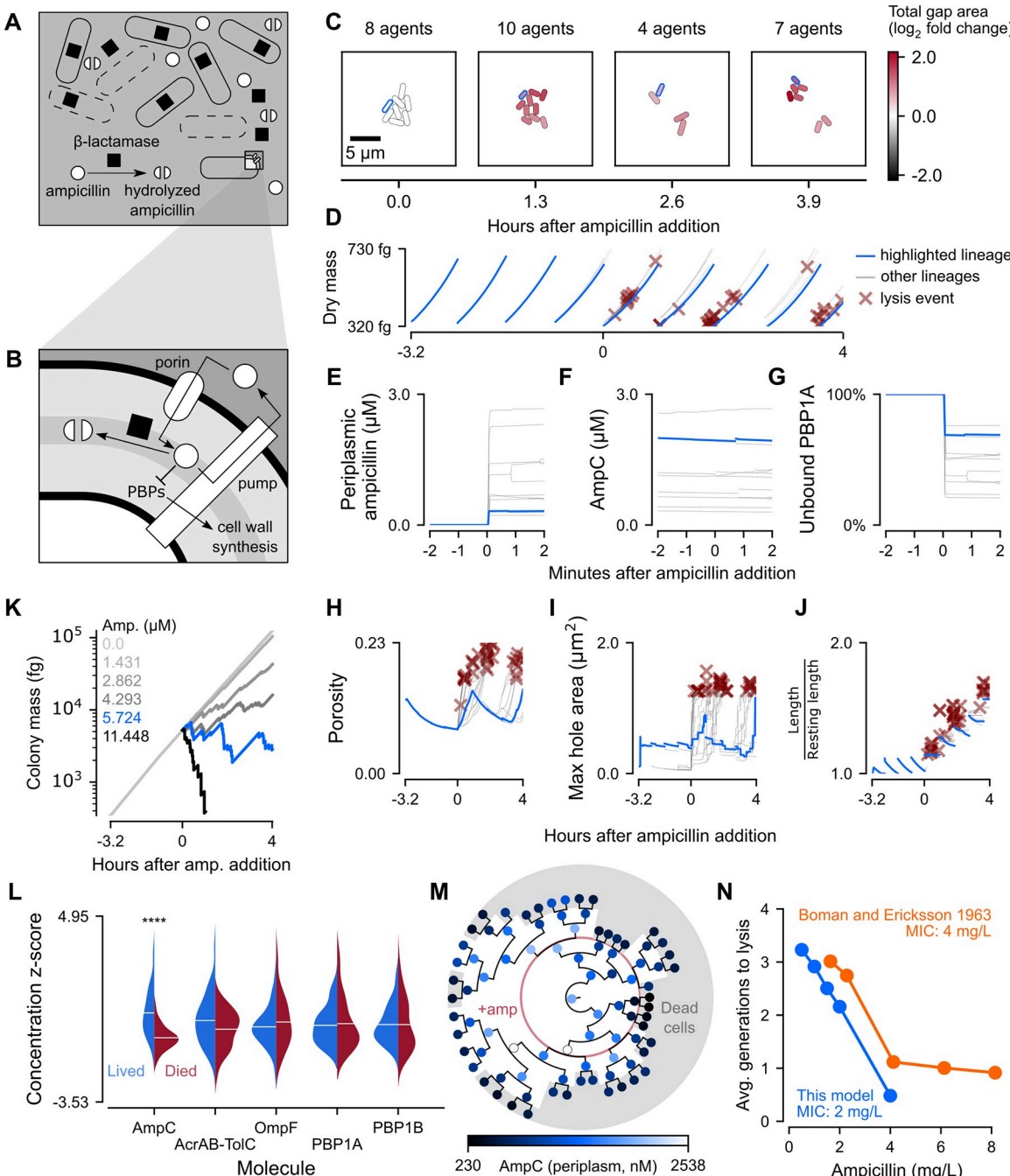

**Fig 4. Spatio-temporal dynamics of response to ampicillin in *E. coli* colonies.** (A-B) Schematic of ampicillin hydrolysis and cell lysis at the colony scale and ampicillin transport and inhibition of cell wall synthesis at the cellular scale. (C) Snapshots of a representative colony simulation after addition of ampicillin at the MIC (2 mg/L). The color of each cell represents the log₂ fold change of total gap area (area not covered by peptidoglycan) compared to the average total gap area for cells in a baseline glucose simulation. The cell lineage ending in cell 001111111 was outlined in blue and also plotted as blue traces in Panels D-J. (D) Per-cell dry mass starting with a single cell grown in M9 minimal medium with 1 mM glucose and no ampicillin for about 3.2 hours (11550 seconds), at which point ampicillin was added at the MIC and the simulation continued for approximately 4 more hours (14550 seconds). (E-G) Per-cell concentration of, from left to right, periplasmic ampicillin, AmpC beta-lactamase, and fraction of active PBP1A in the minutes surrounding tetracycline addition. (H-J) Per-cell concentration of, from left to right, cell wall porosity, maximum hole area, and cell wall stretch for the entire 7.2-hour (26000 seconds) simulation. (K) Whole colony mass for representative simulations run across six different concentrations of ampicillin (MIC in blue). The data from these simulations was also used to create Panel N. (L) Per-cell average concentrations of AmpC, AcrAB-TolC, OmpF, and PBP1A/B were normalized as z-scores and plotted as juxtaposed kernel density estimates, with red representing cells that died and blue representing cells that survived. Stars indicate statistical significance (p << 0.01). (M) Circular

phylogenetic tree of cells colored by AmpC beta-lactamase concentration. Cells in the gray regions died and the red ring represents the time of ampicillin addition. (N) Average number of generations that dying cells were able to survive before lysis in five different concentrations of ampicillin (blue). Number of generations elapsed before seeing a decrease in optical density of a culture across several ampicillin concentrations (orange) [91]. *Note*: All data shown is from simulations run with seed 0 initialized with a snapshot 3.2 hours (11550 seconds) into a baseline glucose simulation (also seed 0) before being continued for an additional four hours (14550 seconds) with varying levels of external ampicillin.

surface area covered by one peptidoglycan unit, and were either occupied by cross-linked peptidoglycan or were considered "gaps". Peptidoglycan strands in the *E. coli* cell wall run mostly parallel, with the glycan backbone oriented along the circumference of the cell [81,84]. We thus modeled cell wall synthesis by PBP1A and PBP1B as the insertion of columns into the lattice, where columns were populated by cross-linked peptidoglycan strands with lengths sampled from a geometric distribution fitted to experimental data (S9A Fig) [85]. At each timestep, we scaled the number of nascent peptidoglycans predicted by our metabolic model by a weighted average of the proportions of unbound PBP1A and PBP1B to determine the pool of peptidoglycans that could be cross-linked into the lattice. Deficiencies in this pool such that cell wall synthesis could not keep pace with cell growth resulted in gaps in the lattice.

The cell wall is elastic in the direction of its long axis [86], and is able to expand and contract up to threefold without permanent deformation or rupture [87]. When damage accumulates, the cell wall "cracks" under turgor pressure, followed by outward bulging through the crack and eventual bursting of the inner membrane [76,88,89]. We chose a theoretical upper bound on cell wall cracking time, by permitting the lattice to stretch as needed to avoid cracking, up until either the experimentally determined maximum reversible extension was reached, or the largest hole expanded beyond a critical threshold for lysis predicted in literature [87,89]. After cell wall cracking, we accounted for the time of inner membrane bulging (on the order of 3.2 minutes) by sampling a waiting time from an exponential distribution fitted to experimental data (S9B Fig) [90]. At the end of this delay, the cell was removed from the simulation, and all of its internal ampicillin and beta-lactamase monomers were spilled into the shared spatial environment where the reaction-diffusion system simulated extracellular enzyme activity.

As with tetracycline, we began our ampicillin simulations with snapshots of colonies 3.2 hours (11550 seconds) into baseline glucose simulations. Globally, we noted that exposing these colonies to ampicillin at the MIC (2 mg/L or 5.724 μM) resulted in a marked but uneven increase in the total area of the cell wall not covered by cross-linked peptidoglycan (Fig 4C), followed by substantial cell death (Fig 4D). Cell death was not concentrated at the beginning or end of the simulated ampicillin exposure, but instead scattered relatively evenly throughout. Notably, there was no observable change in the dry mass growth trajectory of cells following antibiotic addition. This indicated that any observed inhibition of colony mass growth must occur primarily through cell death.

On a single-cell level, periplasmic ampicillin levels rapidly reached equilibrium in a matter of seconds. Unlike tetracycline however, the steady-state concentration of ampicillin varied considerably from cell to cell (CV = 0.78 after one minute) (Fig 4E). The simulated average periplasmic ampicillin concentration of 1.27 μM after 15 minutes of exposure to the MIC was close to the 1.7 μM measured in live cells after the same exposure time [77]. This variation was accompanied by high cell-to-cell variation in the sub-generationally expressed AmpC, the beta-lactamase that hydrolyses ampicillin (CV = 0.83 after one minute) (Fig 4F). This variability in periplasmic ampicillin concentration in turn gave rise to broad heterogeneity of observed PBP inhibition (CV = 0.39 for PBP1A active fraction and 0.30 for PBP1B after one minute) (Fig 4G).

Compared to the seconds required for ampicillin and active PBP levels to reach steady state, damage to the cell wall accumulated on a time scale of many minutes to hours (Fig 4H–4J). Nearly all cells that died experienced steep porosity increases and sudden spikes in maximum hole size immediately preceding lysis (Fig 4H and 4I). Interestingly, sharp spikes in both of these measurements did not guarantee death, with some cells (like those of the lineage highlighted in blue) managing to at least temporarily recover. All cells, regardless of fate, exhibited increasingly stretched cell walls as exposure times lengthened (Fig 4J).

To test the sensitivity of the ampicillin response, we simulated colony exposure to four additional concentrations of ampicillin ranging from one-quarter to double the MIC (Fig 4K). Compared to our simulations with varying tetracycline concentrations (Fig 3L), the total colony mass traces following ampicillin exposure were much more jagged, since at higher concentrations of ampicillin, rapid cell death and a smaller colony size caused each individual cell loss to register as a sharp decrease in total mass.

To determine the most important cellular factors related to ampicillin resistance, we examined the expression of five proteins and protein complexes that play critical roles in the ampicillin response (Fig 4L). Using independent two-sided t-tests, we found that AmpC, the endogenous beta-lactamase, was the only protein for which expression in dead cells was significantly different from that in live cells ($p < 10^{-3}$). Given their similarly high between-cell variability (Fig 4E and 4F), AmpC concentration likely influenced cell fate by controlling periplasmic ampicillin concentration, a hypothesis supported by the strong monotonic relationship between per-cell average AmpC and periplasmic ampicillin concentrations (S10 Fig). By contrast, the average expressions of the AcrAB-TolC multidrug efflux pump, the porin OmpF, and PBPs 1A and 1B exhibited no significant differences between cells that lived and those that died, a finding corroborated by the weak or nonexistent linear relationships between these variables and periplasmic ampicillin concentrations (S10 Fig).

We also found that closely related cells, those that had more recent common ancestors, were more likely to have equivalent survival outcomes (i.e. lived or died) than less related cells (Blomberg's K = 2.31, $p = 10^{-5}$). This suggested that there were traits shared by related cells which influenced cellular survival. Indeed, related individuals resembled each other more in AmpC concentration than random individuals (Blomberg's K = 0.708, $p = 3*10^{-5}$) (Fig 4M). Since *ampC* is known to be a sub-generationally expressed gene (Fig 1C), the inheritance of AmpC by daughter cells following rare transcription events may protect these closely related descendants from lysis. As expected, lineages with higher AmpC expression just before ampicillin addition (eight-cell stage just inside the red circle, Fig 4M) were more likely to have descendants which survived until the end of the simulation. Conversely, lineages which had lower levels of AmpC expression both early on and as the simulation progressed (darker circles) were extremely likely to die out within a generation. This reaffirmed our conclusion that sub-generational expression of AmpC underpinned the variability in cell death due to ampicillin exposure.

Lastly, we plotted the average number of generations from antibiotic addition to cell lysis as a function of external ampicillin concentration, enabling a direct comparison of our simulation output to prior measurements of ampicillin-induced cell death (Fig 4N) [91]. The comparison showed good agreement, especially considering that the MIC of our model was lower than that of the experimentally tested strain. Accordingly, our model yielded more rapid lysis at lower concentrations of ampicillin, as would be expected of a more sensitive strain.

In short, our simulations in ampicillin-containing media exhibited a remarkably strong link between the subcellular hydrolysis of ampicillin and the survival of related cells and even whole colonies, mechanistically bridging a broad range of spatial and temporal scales.

## Discussion

By embedding multiple instances of a whole-cell *E. coli* model into a shared spatial environment model, we were, for the first time, able to bring the molecular mechanisms of whole-cell models to the "whole-colony" scale. After validating the model against prior results, we investigated the dynamics of the *E. coli* antibiotic response across multiple spatial and temporal scales. Among the genes that we implemented new functions for, some (e.g. *ompF* and *tolC*) exhibited high, stable expression while others (e.g. *marR* and *ampC*) had spiking expression with high between-cell variability. Spatial clustering of phenotypically similar cells was weak, most likely due to thermal jitter and pushing forces generated by growth of initially adjacent daughter cells. Under exposure to the bacteriostatic antibiotic tetracycline, cells underwent a concentration-dependent decrease in growth rate that did not reach complete abolition of growth even at concentrations beyond the MIC. Despite heterogeneous expression of the OmpF porin and AcrAB-TolC efflux pump, little variation was observed in the equilibration times and steady-state concentrations of periplasmic and cytoplasmic tetracycline. Conversely, stochastic variation in the concentration of the AmpC beta-lactamase contributed greatly to dramatic cellular differences in steady-state periplasmic ampicillin and was easily identified as the most significant factor in determining cell survival.

A few observations are of particular note. First, we were surprised to find that the relative positions of related cells varied greatly from simulation to simulation. This underlying fact, combined with the inherent variation between even sister cells, meant that there was little clustering of phenotypically similar cells in space. This was confirmed by the weak and often statistically insignificant spatial autocorrelations for expression of OmpF, TolC, MarR, and AmpC monomers (S1 Table). In particular, despite the strong phylogenetic signal for AmpC expression, cells that had low AmpC expression did not exhibit a strong tendency to form tight clusters, causing death by ampicillin to occur almost at random throughout our colonies. This finding has interesting implications for microscopy experiments on colonies of this scale, namely because naïvely evaluating spatial autocorrelations from images may not be sufficient to find significant effects of lineage even when they exist. Interestingly, unlike the flattened, overprovisioned microcolonies of our simulations, resource-limited biofilms have been previously predicted to undergo spontaneous spatial patterning that may contribute to cooperative phenotypes [92,93], indicating that there might be adhesion or other interaction effects between cells that were not considered in our model. Such adhesion effects are also crucial in the human gut, where adhesion to mucosal surfaces enables colonization by both commensal and pathogenic bacteria [94].

In our tetracycline simulations, we observed a potential positive feedback loop of growth inhibition through reduction of active machinery at virtually all steps of the central dogma. This raises the question of what mechanism, if any, real *E. coli* cells might use to counteract this cycle of inhibition. One possibility centers around a reversal of guanosine tetraphosphate (ppGpp)-induced downregulation of rRNA and *rpoA/B/C* (which encode RNA polymerase subunits) transcription [95]. While normally used to slow growth during periods of amino acid starvation, synthesis of ppGpp is also known to be inhibited by the binding of tetracycline to ribosomes [96,97]. Reduced ppGpp levels would, in turn, derepress rRNA and RNA polymerase production. While not included in this work, we recently reported on an expanded version of the whole-cell model which includes mechanisms for ppGpp-mediated growth rate control [98]. We are confident that the incorporation of these and other planned mechanisms will allow our model to yield deeper insights into the single-cell and population-scale dynamics of *E. coli*'s response to tetracycline.

In our ampicillin simulations, we did not anticipate that AmpC expression would be the only significant differentiating factor between live and dead cells. In fact, *in vivo* measurements place the enzymatic rate of the AcrAB-TolC efflux pump higher than that of the AmpC beta-lactamase at periplasmic ampicillin concentrations above 0.8 μM [77,99], indicating that efflux should be a more significant contributor to endogenous ampicillin resistance than hydrolysis. However, while there was a significant amount of heterogeneity in AcrAB-TolC expression, it paled in comparison to the variability in AmpC expression. As a result, most cells had more comparable rates of efflux than they did rates of hydrolysis, and these large differences in hydrolytic rate became the basis for natural selection under ampicillin stress. Notably, all AmpC expression in our model was entirely uninduced, making it all the more surprising that a subset of cells were able to survive and sustain colonies for several generations of growth, at concentrations of ampicillin up to and including the MIC. In the future, it would be worth testing experimentally whether the uninduced, sub-generational expression of AmpC is truly sufficient to sustain colony survival indefinitely at various ampicillin concentrations in culture.

Future refinements of our model could proceed in a number of different directions. For tetracycline, our model does not currently represent a complete mechanism of internal accumulation. Prior publications have proposed that differences in pH between extra- and intracellular compartments may drive accumulation several-fold higher than that explained by membrane potential and concentration gradients alone [9,100]. Higher internal concentrations of tetracycline would perhaps strengthen the level of growth inhibition to the point that growth is abolished entirely at the MIC. In experiments with ampicillin, cells that lacked key division machinery lysed via a gradual loss of cell shape instead of the typically observed mid-cell lesions, strongly suggesting that ampicillin-induced death occurs during division [101]. Indeed, ampicillin was found to bind to a division-associated PBP [102] with much higher affinity than most others [74], including PBP1A and PBP1B. Looking further into the future, we hope to eventually incorporate additional mechanisms for the evolution of antibiotic resistance in the form of genetic heterogeneity and horizontal gene transfer between cells [103]. In real *E. coli* cells, increased expression of *acrAB* was correlated with lower expression of the DNA mismatch repair gene *mutS*, allowing for more rapid spontaneous evolution of resistance phenotypes [104]. In our model, accumulation of mutations could be added in a simplified form as small modulations to basal transcription rates upon division. Additionally, as noted by others [105], cells exposed to antibiotics experience significant metabolic changes with many non-trivial effects. As knowledge of antibiotic resistance mechanisms continues to grow, we can envision a time when our model is able to recapitulate these metabolic changes without being explicitly programmed to do so.

Even without these potential improvements, by synthesizing whole-cell modeling and agent-based modeling using Vivarium, this work was able to produce detailed, multi-scale and biologically realistic simulations, integrating molecular mechanisms and parameters at the scale of protein and metabolic networks with cellular mechanisms at the scale of small colonies. Future work might expand this integration both downward, to incorporate knowledge from molecular-scale models of protein structure and dynamics [106], and upward, to the scale of larger populations with multiple microbial species [107,108]. By expanding our spatial environment model, we may be able to study bacterial utilization of mucus in the intestine or simulate experiments in microfluidic devices. The approach we describe here can in principle accommodate more complex cells and more varied microenvironments, and we are hopeful that these ideas can someday be adapted to create increasingly multiscale models of tumors, tissues, blooms and even entire microbiomes.

## Methods

In this work, we took the whole-cell *E. coli* model described in the supplement of Macklin et al. [22], converted it to use the Vivarium interface [24], added new submodels for the response to ampicillin and tetracycline, and nested it in a spatial environment that supports multiple *E. coli* models running in parallel. Here we briefly summarize how the model was adapted, include a detailed list of simulations run, and describe how simulation outputs were processed to create each figure panel. Further details can be found in S1 Text.

### Colony-scale model

To facilitate integration of multiple whole-cell models into a single shared environment, we first converted the original whole-cell model of *E. coli* into a Vivarium-based composite model. We developed unit tests to ensure that the outputs of the converted submodels matched those from the original model with only small allowances for numerical errors, as well as a composite test to compare outputs from both complete models. The scripts for these tests are located in the *migration* folder of the *vivarium-ecoli* repository.

### Simulated experiments

All simulations were run by executing the *ecoli/experiments/tet_amp_sim.py* file with command-line arguments as listed in Table 1. Simulations were run on a Google Compute Engine virtual machine (VM) with 224 CPUs, 896 GB memory, a 500 GB boot disk with the debian-11-bullseye-v20211105 image, and a 250 GB swap file. Simulation data were emitted to a MongoDB server running on a separate Google Compute Engine VM with 16 CPUs, 32 GB memory, a 500 GB boot disk with the debian-10-buster-v20210316 image, and a 2900 GB attached disk for the database. The complete set of simulations analyzed in this paper took about two days of real time to run.

For each of our experimental conditions (baseline glucose, +tetracycline, and +ampicillin), we ran three replicate 7.2-hour simulations each on a different seed (0, 100, and 10000). All simulations started with one cell at the center of a spatial environment. For the tetracycline and ampicillin conditions, we introduced the antibiotics at their respective MICs (1.5 mg/L and 2 mg/L) after 3.2 hours had elapsed by starting each simulation with a saved snapshot of the corresponding glucose simulation. This allowed us to compare the outcomes for baseline and antibiotic simulations for a given initial seed. These nine simulations (three for each condition) are listed under *Baseline glucose simulations*, *Tetracycline MIC simulations*, and *Ampicillin MIC simulations* in Table 1.

Additionally, to evaluate the effect of varying antibiotic concentration on colony growth, we ran simulations at concentrations besides the MIC for both tetracycline and ampicillin. These simulations were set up identically to those described above, except for the concentration of tetracycline (0.5 mg/L, 1 mg/L, 2 mg/L, or 4 mg/L) and ampicillin (0.5 mg/L, 1 mg/L, 1.5 mg/L, 4 mg/L) introduced, and in that we used just one replicate (on seed 0) for each of these alternative concentrations. These eight simulations (4 for each antibiotic) are listed under *Tetracycline sensitivity simulations* and *Ampicillin sensitivity simulations* in Table 1.

Lastly, in the process of fitting an equilibrium binding constant for aminoacylated-tRNAs with ribosomes, we ran 11 simulations across a wide range of tetracycline concentrations that span several orders of magnitude between 0 and 0.05 mM. We adjusted the binding constant until the resulting dose-response curve was comparable to those reported in literature for cell-free systems (S6A Fig). These simulations are listed under *Tetracycline protein synthesis inhibition*.

Note that our simulation was configured to emit data to a separate MongoDB VM with an internet protocol (IP) address specified in *ecoli/composites/ecoli_configs/cloud.json*. Users can

**Table 1. Catalog of all simulations run and their corresponding command-line parameters.**

| Initial seed (-s) | Baseline (-b) | Initial state (-f)* | Initial colony (-i)** | [Tet.] in mM (-t) | [Amp.] in mM (-a) | Run time (-r)*** | Start time (-n) |
|---|---|---|---|---|---|---|---|
| **Baseline glucose simulations** | | | | | | | |
| 0 | Yes | Default | - | 0 | 0 | 26002 | 0 |
| 100 | | | | | | | |
| 10000 | | | | | | | |
| **Tetracycline MIC simulations** | | | | | | | |
| 0 | No | - | 1 | 3.375E-03 | 0 | 14552 | 11550 |
| 100 | | | 2 | | | | |
| 10000 | | | 3 | | | | |
| **Tetracycline sensitivity simulations** | | | | | | | |
| 0 | No | - | 1 | 1.125E-03 | 0 | 14552 | 11550 |
| | | | | 2.25E-03 | | | |
| | | | | 4.5E-03 | | | |
| | | | | 9E-03 | | | |
| **Tetracycline protein synthesis inhibition** | | | | | | | |
| 0 | No | Default | - | 0 | 0 | 500 | 0 |
| | | | | 5E-05 | | | |
| | | | | 1.08E-04 | | | |
| | | | | 2.32E-04 | | | |
| | | | | 5E-04 | | | |
| | | | | 1.08E-03 | | | |
| | | | | 2.32E-03 | | | |
| | | | | 5E-03 | | | |
| | | | | 1.08E-02 | | | |
| | | | | 2.32E-02 | | | |
| | | | | 5E-02 | | | |
| **Ampicillin MIC simulations** | | | | | | | |
| 0 | No | - | 1 | 0 | 5.72E-03 | 14552 | 11550 |
| 100 | | | 2 | | | | |
| 10000 | | | 3 | | | | |
| **Ampicillin sensitivity simulations** | | | | | | | |
| 0 | No | - | 1 | 0 | 1.43E-03 | 14552 | 11550 |
| | | | | | 2.86E-03 | | |
| | | | | | 4.29E-03 | | |
| | | | | | 1.145E-02 | | |

* Default refers to the file data/wcecoli_t0.json

** Each number refers to a saved snapshot from t = 11550 s of a glucose simulation: 1 = seed 0, 2 = seed 100, 3 = seed 10000. The seed of the saved snapshot was matched to the seed of the antibiotic sim.

*** Additional 2 seconds because no data is emitted after the last 2-second timestep.

either change this IP address or execute *ecoli/experiments/tet_amp_sim.py* with the *-l* flag to emit data to a MongoDB server running on the same machine as the simulation.

## Analyses

**Sub-generational gene expression.** In Fig 1B, genes were considered to exhibit sub-generational expression when they were, on average, expressed less than once per cell for all cells in a baseline glucose simulation (seed 10000). All other genes were considered to exhibit

exponential expression. Genes were considered relevant to the antibiotic response if they were listed under the gene ontology term 0046677 on the EcoCyc database [109]. The *ompF* and *ompC* genes were added to this list, because they are known to encode porins important for antibiotic diffusion into the cell [110].

**Average protein and mRNA concentrations.**    In Fig 2C, the plotted literature mRNA concentrations were estimated as the product of reported average mRNA fractions (count of one mRNA divided by total mRNA count) [52] and the average total mRNA count, divided by the average volume for all cells that lived until division in a representative baseline glucose simulation (seed 10000). In Fig 2D, the plotted literature protein concentrations were derived by dividing the measured count of protein monomers per generation [51] by the average volume of cells that lived until division in a representative glucose simulation (seed 10000). In keeping with their localization to the periplasm and outer membrane, concentrations for OmpF, TolC, and AmpC monomers were calculated using periplasmic volume, which we assumed to be a constant 0.2 times total volume [111]. By contrast, the concentrations for all mRNAs and MarR monomers were calculated using cytoplasmic volume, or 0.8 times total volume.

**Instantaneous growth rate.**    In Fig 3D, instantaneous growth rate was calculated as $G = \log_2(m_1/m_0) / \Delta t$, where $m_1$ is the current dry mass, $m_0$ is the dry mass from the previous time step, and $\Delta t$ is the length of a time step (2 seconds). The instantaneous growth rate for the last time step before division was set to 0. The fold change in instantaneous growth rate was calculated as $G_{tet} / G_{glc,avg}$, where $G_{tet}$ is the instantaneous growth rate for each cell in the tetracycline snapshot and $G_{glc,avg}$ is the average instantaneous growth rate across all cells and time points in a baseline glucose simulation (seed 0).

In Fig 3M, doubling rate was calculated for each time step using the same formula as instantaneous growth rate and converted to 1/hr, yielding the number of times that the dry mass of a cell would double if it continued growing at the same instantaneous rate for an hour. On both scatter plots, each point represents the average doubling rates for a cell within the specified time window post-tetracycline addition (e.g. if a cell is spawned just before an hour after tetracycline addition, only instantaneous doubling rates from before the hour are counted towards its average in the left plot). Cells that spent less than 20 seconds within the specified time windows were excluded because they lacked enough data to calculate an accurate average doubling rate.

**Cell wall integrity.**    To confirm that in our model, introducing ampicillin caused an increase in cell wall damage relative to the baseline condition, we calculated the fold change in total gap area as $(P_{amp} * E_{amp}) / (P_{glc,avg} * E_{glc,avg})$, where $P_{amp}$ is the porosity of the ampicillin-exposed cell wall (number of gaps in lattice divided by total number of positions in lattice), $E_{amp}$ is the stretch over resting length of the ampicillin-exposed cell wall, $P_{glc,avg}$ is the average porosity over all cells and time points of a baseline glucose simulation (seed 0), and $E_{glc,avg}$ is the average stretch over resting length over all cells and time points of a baseline glucose simulation (seed 0). This relative change in cell wall integrity was plotted as an instantaneous $\log_2$ fold change using a color scale in the snapshots of Fig 4C.

In Fig 4H–4J, the first ten seconds of data for each cell were not plotted because of a data artifact. In our model, the cell wall submodel only updated the simulation state every ten seconds to reduce computational burden, resulting in a ten-second period after division where cell wall parameters have not been updated to match.

**Evaluating death factors.**    To determine which antibiotic response-relevant proteins were the major determinants of cell survival, we compared the average expression levels of several proteins and protein complexes between cells that lived and cells that died. Cells that were alive at the end of the simulation were excluded because it was unclear if they would have survived until the next generation. We predicted that beta-lactamase, the OmpF porin,

AcrAB-TolC efflux pump, and PBP1A/1B might contribute to cell survival. We conducted two-sided independent t-tests to compare the mean expression between cells that lived and cells that died, using Bonferroni correction to adjust for multiple testing. In Fig 4L, the per-agent average concentrations of each protein or protein complex were first converted to population z-scores to ease visualization, before being split into live and dead populations whose z-score distributions were subsequently plotted.

**Phylogenetic signal.** We used the Python API of the Environment for Tree Exploration (ETE) toolkit [112] to create the phylogenetic tree shown in Fig 4M and export the structure of the tree in the Newick format. This exported file was used by the R package phytools [113] to compute a phylogenetic signal in the form of Blomberg's K [114] for two traits: a binary variable for whether each cell lived or died and a continuous variable representing the average AmpC concentration for each cell. P-values for these K-statistics were estimated using a permutation test with n = 10,000 randomly permuted phylogenies. A statistically significant K indicates that related tree leaves tend to resemble each other more than unrelated leaves [114]. The value of K itself indicates whether related tips of the phylogenetic tree resemble each other more (K > 1) or less (K < 1) than expected under a Brownian motion model of evolution, which represents neutral evolution that typically occurs without selection [114–116].

**Generations to lysis.** The literature data in Fig 4N was manually extracted from Fig 2 of Boman and Eriksson 1963 [91] using a software tool [117]. The raw data was inverted to yield generations to lysis (time to lysis divided by generation time) instead of generation time divided by time to lysis. The same metric was calculated for simulation data as the average time between ampicillin addition and cell lysis for all cells that died.

## Supporting information

**S1 Fig. Glucose uptake and diffusion.** (A) Distribution of average per-cell glucose uptake rates for a baseline glucose simulation (seed 10000). (B) Cross-sectional glucose concentration in the environment at various time points.
(SVG)

**S2 Fig. Simulated and experimental proteome and fluxome.** (A) Comparison of average protein counts (averaged over each cell's lifespan, then averaged over all cells) for a baseline glucose simulation (seed 10000) against a previously published proteome [44]. (B) Comparison of average central carbon metabolism fluxes (averaged over each cell's lifespan, then averaged over all cells) for a baseline glucose simulation (seed 10000) against a previously published fluxome [45].
(SVG)

**S3 Fig. Desynchronization of division.** (A) Distribution of times until division for all cells in a baseline glucose simulation (seed 10000) compared to the measured doubling time of *E. coli* strain B/r [46]. (B) Standard deviation in birth times for each generation in a baseline glucose simulation (seed 10000).
(SVG)

**S4 Fig. Phylogenetic and physical distances between cells.** Boxplots of Euclidean distance between the centers of cell pairs for each level of phylogenetic relatedness, the minimum number of edges that must be traversed between paired cells in the binary phylogenetic tree. Physical and phylogenetic distances were calculated for cells at the final time point (26000 seconds) of a baseline glucose simulation (seed 10000).
(SVG)

**S5 Fig. Protein monomer concentrations were not correlated.** Scatterplots of per-cell average concentrations for all combinations of OmpF, TolC, MarR, or AmpC, with least-squares regression lines and 95% confidence intervals. Spearman r coefficients and Bonferroni-adjusted p-values in the upper-right corner of each plot. Data taken from baseline glucose simulation (seed 10000).
(SVG)

**S6 Fig. Tuning tetracycline binding and gene regulatory parameters.** (A) Percent inhibition of protein synthesis in our model compared to two cell-free experiments [60,118] for a range of tetracycline concentrations (described under *Tetracycline protein synthesis inhibition* in Table 1). (B) Fold change in mRNA counts (after normalizing by housekeeping gene *gapA* mRNA counts to directly compare with data from literature) when colonies were exposed to tetracycline at the MIC in our model (data combined from seed 0, 100, and 10000) and in a prior experiment [64].
(SVG)

**S7 Fig. Increase in cytoplasmic tetracycline over time.** (A) Decrease in outer membrane permeability at the MIC. (B) Increase in cytoplasmic tetracycline concentration for a range of initial external concentrations (same legend as Panel C). (C) Decrease in AcrAB-TolC concentration for a range of initial external concentrations. All over the course of four hours of tetracycline exposure (seed 0).
(SVG)

**S8 Fig. Inhibition of transcription over time.** Decrease in RNAP and RNA concentrations for 3.2 hours before and 4 hours after exposure to tetracycline at the MIC (seed 0).
(SVG)

**S9 Fig. Fitting parameters for ampicillin submodels.** (A) Geometric distribution fitted to experimentally determined peptidoglycan strand lengths. (B) Exponential distribution fitted to measured time between cell bulging and lysis.
(SVG)

**S10 Fig. Effect of proteins on periplasmic ampicillin concentration.** Scatter plots of per-cell average concentrations of AmpC, OmpF, PBP1A, PBP1B, and AcrAB-TolC against average periplasmic ampicillin concentration. Spearman r coefficients and Bonferroni-adjusted p-values in upper-right corners. Data from simulation with ampicillin at the MIC (seed 0) filtered for data after ampicillin addition (t $\geq$ 11550 seconds). Data for cells alive at the final time point was excluded, because it is unknown whether they would have lived or died given more time.
(SVG)

**S1 Table. Spatial autocorrelation analysis.** Moran's I and Bonferroni-adjusted p-values for concentration of OmpF, TolC, MarR, and AmpC monomers at the end (26000 seconds) of three baseline glucose simulations (seed 0, 100, 10000).
(CSV)

**S1 Text. Supplementary information file.** This file contains more detailed descriptions of the new submodels and other changes, including tables of all the new parameters used during model construction.
(PDF)

## Acknowledgments

We thank the Covert laboratory for helpful discussions and final review of this manuscript.

## Author Contributions

**Conceptualization:** Christopher J. Skalnik, Sean Y. Cheah, Mica Y. Yang, Ryan K. Spangler, Eran Agmon, Markus W. Covert.

**Data curation:** Christopher J. Skalnik, Sean Y. Cheah, Mica Y. Yang, Mattheus B. Wolff.

**Formal analysis:** Christopher J. Skalnik, Sean Y. Cheah, Mica Y. Yang, Mattheus B. Wolff, Eran Agmon.

**Funding acquisition:** Mica Y. Yang, Shayn M. Peirce, Eran Agmon, Markus W. Covert.

**Investigation:** Christopher J. Skalnik, Sean Y. Cheah, Mica Y. Yang, Mattheus B. Wolff.

**Methodology:** Christopher J. Skalnik, Sean Y. Cheah, Mica Y. Yang, Mattheus B. Wolff, Ryan K. Spangler, Lee Talman, Shayn M. Peirce, Eran Agmon, Markus W. Covert.

**Project administration:** Eran Agmon, Markus W. Covert.

**Resources:** Markus W. Covert.

**Software:** Christopher J. Skalnik, Sean Y. Cheah, Mica Y. Yang, Mattheus B. Wolff, Ryan K. Spangler, Jerry H. Morrison, Eran Agmon, Markus W. Covert.

**Supervision:** Eran Agmon, Markus W. Covert.

**Validation:** Christopher J. Skalnik, Sean Y. Cheah, Mica Y. Yang, Mattheus B. Wolff, Eran Agmon.

**Visualization:** Christopher J. Skalnik, Sean Y. Cheah, Mica Y. Yang.

**Writing – original draft:** Christopher J. Skalnik, Sean Y. Cheah, Mica Y. Yang, Eran Agmon, Markus W. Covert.

**Writing – review & editing:** Christopher J. Skalnik, Sean Y. Cheah, Mica Y. Yang, Mattheus B. Wolff, Ryan K. Spangler, Lee Talman, Jerry H. Morrison, Shayn M. Peirce, Eran Agmon, Markus W. Covert.

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
