## [Decision Letter · Decision Letter 0]

30 Sep 2021

Dear Dr. Covert,

Thank you very much for submitting your manuscript "Whole-colony modeling of Escherichia coli" for consideration at PLOS Computational Biology.

As with all papers reviewed by the journal, your manuscript was reviewed by members of the editorial board and by several independent reviewers. In light of the reviews (below this email), we would like to invite the resubmission of a significantly-revised version that takes into account the reviewers' comments.

We cannot make any decision about publication until we have seen the revised manuscript and your response to the reviewers' comments. Your revised manuscript is also likely to be sent to reviewers for further evaluation.

Sincerely,

Joerg Stelling

Associate Editor

PLOS Computational Biology

Daniel Beard

Deputy Editor

PLOS Computational Biology

Reviewer's Responses to Questions

**Comments to the Authors:**

Reviewer #1: The authors developed a software, Vivarium, that integrates whole-cell model (WCM) and agent based model (ABM). This is a first agent-based multiscale model of that simulates whole colonies of E. Coli. While whole-cell models focus on single cells, its utility for E. Coli is diminished as many behaviors of E. Coli is understood in context of many interacting as a group, as in biofilms. Thus integrating WCMs and ABMs, the latter can simulate population behaviors that emerge from interactions of individual agents (or cells or bacteria), could address this gap. The authors’ developed a unified model that simulates independent cells interacting in a shared spatial environment and links individual protein expressions to population-level phenotype. The model was used to demonstrate antibiotic resistance and simulated the colony’s response to antibiotics.

In the model, the agents do not interact directly, but rather the interactions are mediated by changes in the environment, notably through multibody physics, including diffusion. The concentrations of molecules exchanged are through the environment as well. Using this model to investigate response to antibiotics, the authors’ model indicate that variation in the expression level of the betalactamase AmpC, and not of the multi-drug efflux pump AcrAB-TolC, was the key mechanistic driver of survival of E. Coli in the presence of nitrocefin.

Overall, this is a novel approach, however, could the authors model a more commonly used antibiotic against E Coli (for example, doxycycline, ciprofloxacin, or nitrofurantoin) for which the antibiotic resistance is well characterized, which could further demonstrate the breadth of this model.

Reviewer #2: This study builds upon the whole cell Ecoli model (WCM) previously built by the authors and expands it to the population scale.

The WCM model simulates individual Ecoli cells with high complexity.

In stark contrast, their approach to model antibiotic response is extremely rudimentary and oversimplified. The insights from their population model are trivial and a direct consequence of their assumptions.

The novel contribution here is the expansion to a whole colony model and application to antibiotic resistance. These ideas are not well developed unfortunately.

Antibiotic resistance involves numerous transcriptional and metabolic effects. While we dont expect all details of antibiotic response to be modeled, well established mechanisms that are critical to antibiotic action should not be ignored. With the whole cell model they were in a good position to simulate the complexity, but unfortunately that's not the case here.

The way the authors model antibiotic treatment is oversimplified. In the model, the drug kills the cells when it reaches an arbitrary threshold chosen by the authors. The primary targets of many of the antibiotics are well established. It should be straightforward to model the drug effect explicitly by inhibiting cell wall enzymes.  Prior studies, for example from the Palsson lab, have used FBA models to simulate inhibition of antibiotic targets. The authors could easily build upon that.

The authors conclude from their modeling that "the AcrAB-TolC concentration had little effect" on antibiotic response. This is incorrect and goes against several known observations. For example, how can they explain the fact that efflux pump inhibitors are synergistic with many antibiotics and are effective? Deletion of acrAB genes greatly increases sensitivity to antibiotics in genome-wide knockout screens. Overall, the model is too simplistic model to make such a claim.

The authors state in the discussion (line 370-74) that many of the kinetic parameters related to beta lactamase and porin transcription, expression, and activity are unknown and could be inaccurate. The authors should report these parameters and do some type of sensitivity analysis of their results. If most parameters are unknown, may be their model is not setup to answer this type of question.

In the abstract and discussion, the authors report that they use a novel software called vivarium, which is misleading, as vivarium is not presented here but is reported in a different related manuscript. The authors cannot claim novelty in two separate papers.

There is no validation per se for the whole colony model. The only 'validation' reported in line 202, which they state "deepened our confidence in the simulations" is trivial. The authors found that the model produced higher growth in aerobic vs anaerobic. This has nothing to do with the colony model. Even a simple FBA model could have done this.

How is AmpC induced and regulated in response to antibiotics? How long is the time delay for induction? It appears to be constitutively expressed. I am surprised ampC is even expressed in their model.  It is usually expressed in clinical strains. The model they are using is not a clinical strain or atleast it wasnt mentioned in the manuscript. AmpC expression is usually low but inducible in response to β-lactam exposure.

What are the induction times for ampc and tolc?

How was protein specific heterogeneity parameters determined in the model?  How/why is Ampc more noisy than Acrab? The reason for this, and if it was intentionally programmed, should be discussed.

is there a cost for producing beta lactamase? if not why is that the case?

I think there would be some type of interaction effect between porins and belactamase activity, but I'm surprised they did not see anything.

Nitrocefin is technically not an antibiotic. it is an indicator dye for betalactamase detection.

Their model currently treats the drug treatment as a bacteriostatic effect, but betalactams are bactericidal. The dead cells should be removed from simulation.

Why do dead cells take up space?

Not sure how they got the MIC value for nitrocefin. The original nitrocefin article reports 64 ug/ml

The authors should take an unbiased approach and compare all the molecular properties of alive and dead cells.

The authors describe the implementation of the physics problem related to forces acting on cells. But no results are described from this analysis. What did we learn from the physics problem?

The authors refer to the antibiotic resistance regulator, marA, in the introduction and discussion but never use it in the paper.

line 120: Which molecules from WCM are reported to vivarium?  Why not all?  What is the rationale for selecting a small set of genes/features?

line 178: The authors report variation in glucose levels. What about other nutrients and secreted metabolites? do they change? that could be reported in a heatmap and correlated with antibiotic response.

**Have the authors made all data and (if applicable) computational code underlying the findings in their manuscript fully available?**

Reviewer #1: Yes

Reviewer #2: Yes

PLOS authors have the option to publish the peer review history of their article (what does this mean?). If published, this will include your full peer review and any attached files.

Reviewer #1: No

Reviewer #2: No
---

## [Decision Letter · Decision Letter 1]

1 Jun 2023

Dear Dr. Covert,

We are pleased to inform you that your manuscript 'Whole-cell modeling of E. coli colonies enables quantification of single-cell heterogeneity in antibiotic responses' has been provisionally accepted for publication in PLOS Computational Biology.

Best regards,

Daniel A Beard

Section Editor

PLOS Computational Biology

Daniel Beard

Section Editor

PLOS Computational Biology

Reviewer's Responses to Questions

**Comments to the Authors:**

Reviewer #1: The authors have sufficiently addressed all comments.

Reviewer #2: The authors have done an excellent job addressing comments from prior round of review.

**Have the authors made all data and (if applicable) computational code underlying the findings in their manuscript fully available?**

Reviewer #1: None

Reviewer #2: Yes

PLOS authors have the option to publish the peer review history of their article (what does this mean?). If published, this will include your full peer review and any attached files.

Reviewer #1: No

Reviewer #2: No

---

## [Editor Report · Acceptance letter]

13 Jun 2023

PCOMPBIOL-D-21-00770R1 

Whole-cell modeling of E. coli colonies enables quantification of single-cell heterogeneity in antibiotic responses

Dear Dr Covert,

I am pleased to inform you that your manuscript has been formally accepted for publication in PLOS Computational Biology. Your manuscript is now with our production department and you will be notified of the publication date in due course.

With kind regards,

Zsofia Freund
